

# Geostrophic adjustment on the mid-latitude $\beta$-plane

Itamar Yacoby, Nathan Paldor, and Hezi Gildor

Fredy and Nadine Herrmann Institute of Earth Sciences, Edmond J. Safra Campus, Givat Ram, the Hebrew University of Jerusalem, Jerusalem, Israel

**Correspondence:** Nathan Paldor (nathan.paldor@mail.huji.ac.il)

**Abstract.** Analytical and numerical solutions of the Linearized Rotating Shallow Water Equations are combined to study the geostrophic adjustment on the mid-latitude $\beta$-plane. The adjustment is examined in zonal periodic channels of width $L_y = 4R_d$ ('narrow' channel, where $R_d$ is the radius of deformation) and $L_y = 60R_d$ ('wide' channel) for the particular initial conditions of a resting fluid with a step-like height distribution, $\eta_0$. In the one-dimensional case, where $\eta_0 = \eta_0(y)$ we find that: (i) $\beta$ affects

the geostrophic state (determined from the conservation of the meridional vorticity gradient) only when $b = \cot(\phi_0)\frac{R_d}{R} \geq 0.5$ (where $\phi_0$ is the channel's central latitude and $R$ is Earth's radius); (ii) The energy conversion ratio varies by less than $10\%$ when $b$ increases from 0 to 1; (iii) In 'wide' channels, $\beta$ affects the waves significantly even for small $b$ (e.g. $b = 0.005$). (iv) For $b = 0.005$, harmonic waves approximate the waves in 'narrow' channels, and trapped waves approximate the waves in 'wide' channels. In the two-dimensional case, where $\eta_0 = \eta_0(x)$ we find that: (i) At short times the spatial structure of the

steady solution is similar to that on the $f$-plane, while at long times the steady state drifts westward at the speed of Rossby waves — harmonic Rossby waves in 'narrow' channels and trapped Rossby waves in 'wide' channels; (ii) In 'wide' channels, trapped wave dispersion causes the equatorward segment of the wavefront to move faster than the northern segment; (iii) The energy of Rossby waves on the $\beta$-plane approaches that of the steady-state on the $f$-plane; (iv) The results outlined in (iii) and (iv) of the one-dimensional case also hold in the two-dimensional case.

## 1 Introduction

It has long been established that large scale flows in the ocean and atmosphere are in near geostrophic balance whereby the pressure gradient force is balanced by the Coriolis force [see, e.g., Gill (1982, section 7.6), and Vallis (2017, chapter 5)]. The fundamental theory of the way in which these flows transition from an initial unbalanced state to a geostrophically balanced state (known as Geostrophic Adjustment) is a corner-stone of geophysical fluid dynamics as it is crucial for understanding the

dynamics of the ocean and the atmosphere (see e.g., Gill, 1982). Despite decades of research, the undersnading of geostrophic adjustment is far from being complete.

The geostrophic adjustment theory was first studied in the 1930s when Carl-Gustaf Rossby published the pioneering works on the subject (Rossby, 1937, 1938). Rossby's work was extended in several papers, e.g., Blumen (1972) and Yacoby et al. (2021). However, as in Rossby's original studies, nearly all of these earlier studies addressed particular aspects (e.g. initial

or boundary conditions) of the adjustment on the $f$-plane; thus, their applicability to the real ocean is limited. Though the adjustment theory was extended to the equatorial $\beta$-plane (section 11.11 in Gill, 1982; Killworth, 1991; Rostami and Zeitlin,



2019, 2020) and to the sphere (Paldor and Dritschel, 2021), only marginal advances were made in extending the geostrophic adjustment theory to mid-latitude $\beta$-plane. In his discussion of the expected effect of $\beta$ on the adjustment in mid-latitudes, Blumen (1972, section C2) commented that on the $\beta$-plane the fluid should adjust to the quasi-geostrophic Rossby waves in

the way it adjusts to the steady geostrophic state on the $f$-plane. Similarly, in their study of the non-linear adjustment process on the $f$-plane, Reznik et al. (2001, section 5) commented heuristically on the changes in the theory that should be expected when $\beta \neq 0$, but have not developed a complete theory.

The waves that develop on the $f$- and $\beta$-planes during the adjustment process are determined by the boundary conditions and by the governing equations as follows: (i) In unbounded domains on the $f$-plane, only Poincaré waves develop (Cahn, 1945);

(ii) In channels on the $f$-plane, Kelvin and Poincaré waves develop (Gill, 1976, 1982); (iii) In unbounded domains on the mid-latitude $\beta$-plane, Rossby and Poincaré waves develop (Blumen, 1972; Gill, 1982); (iv) On the equatorial $\beta$-plane, Rossby, Kelvin and Poincaré waves develop (Gill, 1982; Rostami and Zeitlin, 2019); (v) In channels on the mid-latitude $\beta$-plane, Rossby, Kelvin and Poincaré waves develop as in (iv) [follows from the combination of (ii) and (iii)].

In wide channels on the mid-latitude $\beta$-plane, an alternate theory, the trapped wave theory, was developed for Poincaré and

Rossby waves (Paldor et al., 2007; Paldor and Sigalov, 2008; Paldor, 2015, see details in §3 below). The relevance of the trapped wave theory was recently confirmed using numerical simulations (Gildor et al., 2016) and satellite observations from the Indian Ocean (De-Leon and Paldor, 2017). The trapped waves theory is employed in this study as it underscores the effect of $\beta$ on waves in wide channels (see §4.3, §5.2-5.3, and §6.2)

This paper examines the geostrophic adjustment theory in periodic zonal channels on the mid-latitude $\beta$-plane and its outline

is: In §2 the governing equations, the numerical schemes used and the set-up of the problem are presented. In §3 we briefly compare the harmonic wave theory and the trapped wave theory and classify the width of the channels in which each of these theories is expected to be valid. For each theory, we summarize the analytical expressions for the dispersion relations and the eigenfunctions of both Poincaré and Rossby waves. In §4 we address the one-dimensional zonally-invariant adjustment problem when the initial height distribution, $\eta_0$, is a function of $y$. In §5 we address the two-dimensional adjustment problem

when $\eta_0$ is a function of $x$ (the $y$ dependence is explicit in $f(y)$). In both the one-dimensional and the two-dimensional cases, we examine the effect of $\beta$ on the: (i) Geostrophic steady state (§4.1 and §5.1); (ii) Waves' structure and spectrum (§4.3 and §5.2-5.3); (iii) Energetics of the adjustment process (§4.2 and §5.4). The paper ends with a discussion and summary in §6.





## 2 Set-up of the problem

### 2.1 Governing equations

The (inviscid) Linearized Rotating Shallow Water Equations (LRSWE) are:

$$\frac{\partial u}{\partial t} - f(y)v = -g\frac{\partial \eta}{\partial x}, \tag{1}$$

$$\frac{\partial v}{\partial t} + f(y)u = -g\frac{\partial \eta}{\partial y}, \tag{2}$$

$$\frac{\partial \eta}{\partial t} + H\left(\frac{\partial u}{\partial x} + \frac{\partial v}{\partial y}\right) = 0, \tag{3}$$

where $u$ and $v$ are the velocity components along the $x$ (zonal) and $y$ (meridional) coordinates, respectively, $\eta$ is the deviation

of the fluid height from its mean value $H$, and $g$ is the gravitational acceleration (or the reduced gravitational acceleration when

the fluid is stratified). On the mid-latitude $\beta$-plane the Coriolis frequency, $f(y) = 2\Omega \sin(\phi)$ (where $\Omega$ is Earth's frequency of

rotation and $\phi$ is the latitude), is expanded to first order in $y$ about a mean latitude $\phi_0$ (where $y$=0), i.e.,

$$f(y) = f_0 + \beta y = 2\Omega\left(\sin(\phi_0) + \frac{\cos(\phi_0)}{R}y\right)$$

where $R$, as mentioned above, is Earth's mean radius.

To reduce the number of free parameters in the problem, we scale $t$ on $f_0^{-1}$, $(x,y)$ on the radius of deformation – $R_d = \sqrt{gH}/f_0$, $\eta$ on the initial disturbance amplitude $\tilde{\eta}$, and $(u,v)$ on $\tilde{\eta}\sqrt{gH}/H$. This scaling guarantees that the amplitude of

$\eta(t=0)$ is 1 and yields the non-dimensional equations:

$$\frac{\partial u}{\partial t} - (1+by)v = -\frac{\partial \eta}{\partial x}, \tag{4}$$

$$\frac{\partial v}{\partial t} + (1+by)u = -\frac{\partial \eta}{\partial y}, \tag{5}$$

$$\frac{\partial \eta}{\partial t} + \frac{\partial u}{\partial x} + \frac{\partial v}{\partial y} = 0. \tag{6}$$

This system contains the single free parameter (the "non-dimensional $\beta$"):

$$b = \frac{\beta R_d}{f_0} = \cot(\phi_0)\frac{R_d}{R}.$$

It should be noted that, formally, this scaling applies to the northern hemisphere where $\phi_0 > 0$ i.e. $f_0 > 0$. In the southern

hemisphere, a minus sign should be added to the scales of $x$, $y$ (i.e. to $R_d$) and $t$ while the parameter $b$ and the variables $u$, $v$

and $\eta$ are unchanged.

Subtracting the $y$ derivative of (4) from the $x$ derivative of (5) yields the vorticity equation:

$$\frac{\partial}{\partial t}\left(\frac{\partial v}{\partial x} - \frac{\partial u}{\partial y}\right) + (1+by)\left(\frac{\partial u}{\partial x} + \frac{\partial v}{\partial y}\right) = -bv,$$

which together with the continuity equation, (6), yields:

$$\frac{\partial}{\partial t}\left(\frac{\partial v}{\partial x} - \frac{\partial u}{\partial y} - (1+by)\eta\right) = -bv. \tag{7}$$

This equation is used in §4.1 to find the geostrophic steady state.



**Table 1.** The numerical simulations and their basic parameters as defined in §2.2-2.3.

| Simulation | Model | $\Delta t$ | $\Delta x$ | $\Delta y$ | $L_x$ | $t_{end}$[a] | Sampling intervals | Figure |
|---|---|---|---|---|---|---|---|---|
| **A** | RSW solver | $10^{-3}$ | $L_x/4$[b] | 0.01 | $4\Delta x$[b] | 200 | 0.1 | 2 |
| **B** | RSW solver | $10^{-3}$ | $L_x/4$ | Wide[c]: 0.1 Narrow: 0.01 | $4\Delta x$ | $10^3$ | 0.1 | 3-8 |
| **C** | RSW solver | $10^{-2}$ | Wide: 0.2 Narrow: 0.1 | Wide: 0.2 Narrow: 0.1 | 120 | $10^5$ | 20 | 10-11,16 |
| **D** | MITgcm | $10^{-1}$ | 2 | Wide: 1 Narrow: 0.1 | 120 | $10^6$ | 100 | 12-13 |
| **E** | RSW solver | $10^{-3}$ | 0.1 | Wide: 0.1 Narrow: 0.01 | 12 | $10^3$ | 0.1 | 14-16 |

[a]Run time

[b] I.e., the number of cells in the zonal direction is 4 — the minimal value allowed by the solver (see §2.3).

[c] 'Wide' and 'Narrow' refer to wide channel ($L_y = 60$) and narrow channel ($L_y = 4$), respectively.

## 2.2 Numerical scheme

The time-dependent system (4)-(6) is solved numerically using the Rotating Shallow Water (RSW) solver described in Gildor et al. (2016). Briefly, the finite difference numerical scheme employs a leapfrog time-differencing and central spatial differencing on an Arakawa $C$ grid. A Robert-Asselin filter (Haltiner and Williams, 1980) is applied to suppress numerical modes.
More details on this solver can be found in Gildor et al. (2016). The time step ($\Delta t$) and the grid spacing ($\Delta x$, $\Delta y$) were varied and are noted in table 1 for each simulation (indicated in the table by letters **A-E**). For several simulations, the results of the RSW calculations were compared to those obtained with the Massachusetts Institute of Technology General Circulation Model (MITgcm, see Marshall et al., 1997). In all such comparisons, the results obtained with the two models were identical and for the most part, the RSW solver was less stable than the MITgcm and required a smaller $\Delta t$ for given $\Delta x$ and $\Delta y$ so it required
a significantly longer computation time. Therefore, the longest simulation (**D**, see $t_{end}$ in table 1) was carried out using the MITgcm.





## 2.3 Channel configurations and boundary conditions

The adjustment is studied in a periodic zonal channel (i.e., where the channel walls of length $L_x$ are aligned along the $x$-direction and the channel width is $L_y$. When examining the one-dimensional ($x$-independent) problem (see §4) the number of

cells in the zonal direction was set to the minimal value allowed by the solver: $L_x/\Delta x = 4$. As expected, no variations in the zonal direction were developed in the numerical solution. When examining the two-dimensional problem (see §5) the channel length was varied from $L_x = 12$ to $L_x = 120$ (the length is noted in each case). The channel width was varied from $L_y = 4$ in narrow channels to $L_y = 60$ in wide channels. The origin of the $y$-coordinate is chosen such that the channel walls are located at $y = \pm\frac{L_y}{2}$. The boundary conditions at the channel's meridional boundaries are the vanishing of the normal velocities, i.e.:

$$v = 0 \quad \text{at} \quad y = \pm\frac{L_y}{2}. \tag{8}$$

The boundary conditions at the channel's zonal boundaries are periodicity of $u$, $v$ and $\eta$.

## 2.4 Initial conditions

Throughout this work the fluid is assumed to be initially at rest, i.e. (the subscript 0 in $u$, $v$ and $\eta$ denotes initial values),

$$u_0 = v_0 = 0. \tag{9}$$

In the one-dimensional case (§4) the initial surface height distribution, $\eta_0$ is a function of $y$ given by:

$$\eta_0(y) = \text{sgn}(y), \tag{10}$$

while in the two-dimensional case (§5) $\eta_0$ is a function of $x$ given by

$$\eta_0(x) = \begin{cases} 0, & \text{for } |x| > D, \\ 1, & \text{for } |x| < D, \end{cases} \tag{11}$$

with $D = L_x/12$. An initial step condition, $\eta_0(x) = \text{sgn}(x)$, is not addressed here since it violates the assumed periodicity in

the boundary conditions.

## 3 Harmonic- and trapped-wave theories in a channel

We now let the $(u, v)$ velocity components and $\eta$ in the LRSWE (4)-(6) vary with $x$ and $t$ as a zonally travelling wave with $y$-dependent amplitude, namely:

$$(u, v, \eta) = \big(\hat{u}(y), \hat{v}(y), \hat{\eta}(y)\big)e^{i(kx-\omega t)}$$

where $k$ is the wave's zonal wavenumber and $\omega$ is its frequency. As was shown in Paldor et al. (2007) and Paldor and Sigalov (2008), substituting these expressions for $u$, $v$ and $\eta$ in equations (4)-(6), eliminating $\hat{u}$ and $\hat{\eta}$ and neglecting second-order



terms in $by$ [i.e., the term $b^2y^2\hat{v}$, noting that second-order terms in $y$ have already been neglected in the $1^{st}$ order expansion of $f(y)$] yields the following Schrödinger eigenvalue equation for $\hat{v}$:

$$\frac{\partial^2 \hat{v}}{\partial y^2} + (E - 2by)\hat{v} = 0 \tag{12}$$

where $\hat{v}$ is the eigenfunction and

$$E = \omega^2 - k^2 - \frac{bk}{\omega} - 1 \tag{13}$$

is the eigenvalue. In the high-frequency limit, $\frac{bk}{\omega} \ll 1$ and can be neglected, which yields the approximate dispersion relation for Poincaré waves

$$\omega^2 = 1 + k^2 + E. \tag{14}$$

In the low-frequency limit, $\omega^2$ can be neglected which yields the approximate dispersion relation for Rossby waves:

$$\omega = \frac{-bk}{1 + k^2 + E}. \tag{15}$$

The value of $E$, determined from the solution of the eigenvalue equation (12), varies with $b$: for small (respectively, large) value of $b$ it yields harmonic (respectively, trapped) waves.

For completeness, in the following subsections we briefly outline the main properties of these two wave types and delineate

values of $b$ and $L_y$ where each of them is expected to prevail.

## 3.1   Harmonic waves

In the harmonic theory the $y$-dependent term, $-2by$, is neglected in (12) which is solved by harmonic eigenfunctions:

$$\hat{v}_n = \sin\left[\frac{\pi(n+1)}{L_y}\left(y + \frac{L_y}{2}\right)\right] \tag{16}$$

with the corresponding eigenvalues:

$$E_n = \left(\frac{\pi(n+1)}{L_y}\right)^2. \tag{17}$$

Substituting the expression for $E_n$ found above in the dispersion relations (14)-(15) yields:

$$\omega^2 = 1 + k^2 + \left(\frac{\pi(n+1)}{L_y}\right)^2, \quad \omega = \frac{-bk}{1 + k^2 + \left(\frac{\pi(n+1)}{L_y}\right)^2} \tag{18a,b}$$

for harmonic Poincaré waves and harmonic Rossby waves, respectively.

## 3.2   Trapped waves

In the trapped wave theory, the term $-2by$ in equation (12) is retained. Defining

$$z(y) = -(2b)^{-2/3}\big[E - 2by\big],$$



transforms equation (12) to the Airy equation:

$$\frac{\mathrm{d}^2 \hat{v}}{\mathrm{d}z^2} - z\hat{v} = 0 \tag{19}$$

The general solution of (19) is a linear combination of $Ai(z)$, that decays (faster than exponential) for $z > 0$, and $Bi(z)$, that grows (faster than exponential) for $z > 0$. When $z\left(y = +\frac{L_y}{2}\right)$ is sufficiently large, $Ai\left[z\left(y = +\frac{L_y}{2}\right)\right]$ is negligibly small so this function alone nearly satisfies the boundary condition at the northern wall, $y = +\frac{L_y}{2}$ so the contribution of $Bi(z)$ to the general solution has to be negligible. For example, for $z\left(y = +\frac{L_y}{2}\right) = 2$ the value of Ai at the northern wall is $Ai\left[z\left(y = +\frac{L_y}{2}\right)\right] \leq$ 0.035 while the value of $Bi(2) \approx 3.3$ there. Thus, the amplitude of $Bi(z)$ in the general solution (that consists of a linear combination of Ai and Bi) is two orders of magnitude smaller than that of $Ai(z)$.

The boundary condition at the southern wall, $y = -\frac{L_y}{2}$, is satisfied by setting $z\left(y = -\frac{L_y}{2}\right)$ to the $n$th zero of $Ai(z)$ (denoted as $-\xi_n$ with $\xi_n > 0$ since all zeros of $Ai(z)$ are negative) and this condition determines the discrete eigenfunctions:

$$\hat{v}_n = Ai\left[(2b)^{1/3}\left(y + \frac{L_y}{2}\right) - \xi_n\right] \tag{20}$$

with the corresponding eigenvalues:

$$E_n = (2b)^{2/3}\xi_n - bL_y. \tag{21}$$

Substituting this expression for $E_n$ in the dispersion relations (14)-(15) yields:

$$\omega^2 = 1 + k^2 + (2b)^{2/3}\xi_n - bL_y \tag{22}$$

for trapped Poincaré waves, and

$$\omega = \frac{-bk}{1 + k^2 + (2b)^{2/3}\xi_n - bL_y} \tag{23}$$

for trapped Rossby waves.

As was shown in Paldor and Sigalov (2008) and Gildor et al. (2016), the harmonic-wave theory provides accurate approximations for waves in narrow channels while the trapped-wave theory does so in wide channels. The wide channel scenario applies when $z\left(y = +\frac{L}{2}\right)$ is increased above a threshold value, $Z^*$, namely, when:

$$z\left(y = +\frac{L_y}{2}\right) = -(2b)^{-2/3}\left[E - bL_y\right] > Z^* \tag{}$$

where $Z^*$ is sufficiently large. Using the expression for $E_n$, (21), leads to the following constraint on $L_y$:

$$L_y > (2b)^{-\frac{1}{3}}\left(Z^* + \xi_n\right) \tag{24}$$

which indicates that the higher the wave mode, $n$ (and with it, value of $\xi_n$), the wider the channel should be for the trapped wave theory to hold. As mentioned above, $Z^*$ should be a "sufficiently large" number and following the discussion in the beginning of this subsection $Z^* = 2$ should be considered "sufficiently large". We examine this subtle issue in §4.3 [see equation (36)].

Having established analytically the character of the transient waves in wide and narrow channels we now turn to numerical solutions of the adjustment problem in one- and two-dimensions.





## 4 One-dimensional adjustment problem

When $\eta_0$ is a function of $x$ only, both the initial conditions (9)-(10) and the coefficients of the LRSWE are independent of $x$ so the solutions can be assumed independent of $x$ at all time. In this case, the governing equations (4) - (6) reduce to:

$$\frac{\partial u}{\partial t} - (1 + by)v = 0, \tag{25}$$

$$\frac{\partial v}{\partial t} + (1 + by)u = -\frac{\partial \eta}{\partial y}, \tag{26}$$

$$\frac{\partial \eta}{\partial t} + \frac{\partial v}{\partial y} = 0, \tag{27}$$

and equation (7) becomes:

$$\frac{\partial}{\partial t}\left(\frac{\partial u}{\partial y} + (1 + by)\eta\right) = bv. \tag{28}$$

Substituting the continuity equation, (27), in the $y$ derivative of (28) yields:

$$\frac{\partial}{\partial t}\left(\frac{\partial^2 u}{\partial y^2} + (1 + by)\frac{\partial \eta}{\partial y} + 2b\eta\right) = 0. \tag{29}$$

This conservation equation for the meridional vorticity gradient (plus $b\eta$ that originates from the $y$-derivative of $bv$ on the

RHS of (28)) implies that the combination of time-dependent variables in the bracket at time $t$ equals their initial combination (denoted by the subscript "0") i.e.:

$$\frac{\partial^2 u}{\partial y^2} + (1 + by)\frac{\partial \eta}{\partial y} + 2b\eta = \frac{\partial^2 u_0}{\partial y^2} + (1 + by)\frac{\partial \eta_0}{\partial y} + 2b\eta_0. \tag{30}$$

In contrast to the $f$-plane, where the vorticity is conserved, on the $\beta$-plane the conserved quantity is the *meridional gradient* of the vorticity. Thus, while on the $f$-plane (28) with $b = 0$ yields the relation between the initial and final states, on the $\beta$-plane,

this relation is derived from the $y$-derivative of the vorticity equation — equation (30). Indeed, on the $f$-plane the vorticity conservation was employed in Gill (1976, equations (4.7)-(4.8)) and Gill (1982, equations (7.2.8)-(7.2.10)).

### 4.1 Steady state

The steady (geostrophic) solution of equations (25) - (26) (denoted by the subscript "g") satisfies:

$$v_g = 0 \tag{31}$$

and

$$\frac{\partial \eta_g}{\partial y} = -(1 + by)u_g. \tag{32}$$

No information beyond this geostrophic relation between $u_g$ and $\eta_g$ is contained in system (25) - (27) when $\frac{\partial}{\partial t} = 0 = v$. However, a second general relation between $u$ and $\eta$ is provided by equation (30) that expresses the conservation of the





meridional gradient of vorticity and this general relation applies also to the steady solutions. The geostrophic relation can
be combined with the conservation of meridional vorticity gradient to uniquely determine $u_g$ and $\eta_g$. The system (30) and
(32) can only be solved numerically by imposing relevant boundary conditions and solving the associated eigenvalue problem
(we used Matlab solver bvp4c). The boundary conditions for $u$ were derived as follows: Substituting the boundary condition
$v\left(\pm\frac{L_y}{2}\right) = 0$ in equation (25) yields $\frac{\partial u}{\partial t}\big|_{y=\pm L_y/2}$ so $u$ retains its initial value at the boundaries i.e.:

$$u = u_0 = 0 \quad \text{at} \quad y = \pm\frac{L_y}{2}. \tag{33}$$

Note that formally, condition (33) determines the vanishing of $u$ at the boundaries and not the vanishing of the steady
geostrophic field $u_g$. However, since $u_g$ is the averaged distribution (over many wave-periods) of $u$, $u_g$ must also satisfy
condition (33). The boundary conditions for $\eta_g$ were determined as follows: for large channels ($L_y = 60$) we assume that the
geostrophic state is confined to a relatively small area around $x = 0$ so:

$$\eta_g = \eta_0 = \pm 1 \quad \text{at} \quad y = \pm\frac{L_y}{2}. \tag{34}$$

For narrow channels ($L_y = 4$) we impose the conditions:

$$\frac{\partial \eta_g}{\partial y} = 0 \quad \text{and} \quad \eta_g = -a \quad \text{at} \quad y = -\frac{L_y}{2} \tag{35}$$

where the value of $a$ is determined by mass conservation:

$$\int_{-L_y/2}^{L_y/2} \eta_g \mathrm{d}y = \int_{-L_y/2}^{L_y/2y} \eta_0 \mathrm{d}y = 0,$$

i.e., we choose for $a$ the value that minimizes the integral:

$$\int_{-L_y/2}^{L_y/2} \eta_g \mathrm{d}y.$$

Figure 1 compares the geostrophic solutions on the $\beta$-plane (solid lines) and the $f$-plane (dashed lines). In mid-latitudes
($\pi_0 = \pi/4$), the value $b = 0.005$ corresponds to $R_d \sim 30\ km$ – a typical oceanic value whereas, $b = 0.05$ is corresponds to
$R_d \sim 300\ km$ – a typical atmospheric value. The figure clearly shows that the effect of $\beta$ on the geostrophic state is relatively
small and becomes significant only at large $b$-values that are unacceptable for mid-latitude $\beta$-plane since $bL_y/2 \geq 1$ (i.e.,
$\beta L_y/2 \geq f_0$ in dimensional notation). Such unacceptable cases are evident in figure 1 in the $b = 0.05$ row of a wide channel
(left column, middle panel – $bL_y/2 = 1.5$) and both columns of the $b = 0.5$ row (bottom panels – $bL_y/2 = 15$ in the left panel
and $bL_y/2 = 1$ in the right panel). Thus, in all acceptable values of $b$ and $L_y$ the steady state on the $f$-plane provides an
accurate estimate of the same state on the $\beta$-plane.

### 4.2 Energy

Having analyzed the effect of $\beta$ on the steady geostrophic state, we turn to the examination of its effect on the energetics of
the adjustment process. This analysis proved to be very informative on the $f$-plane where it quantifies the division between the





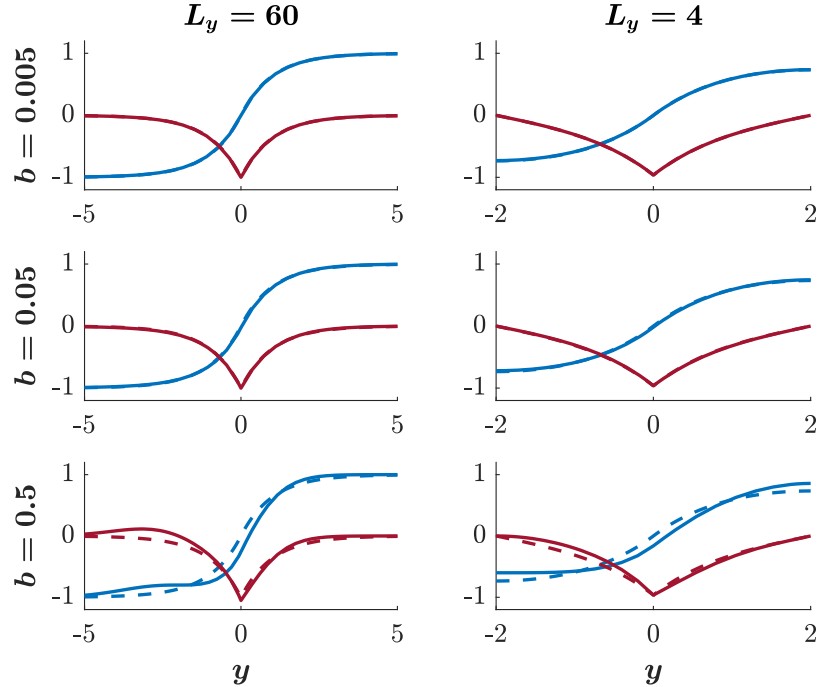

**Figure 1.** The spatial structure of the geostrophic steady state, $\eta_g$ (blue) and $u_g$ (red). Solid lines: the distributions for indicated values of $b$; Dashed lines: the distributions for $b = 0$ (i.e. on the $f$-plane). Left column: A wide channel; Right column: A narrow channel. In all steady states $v = 0$. At smaller values of $b$ the solid lines coincide with the dashed lines. For the $f$-plane we use the analytic solutions (A31)-(A32) of Yacoby et al. (2021) with $\eta_g(x)$ replaced by $\eta_g(y)$ and $v_g(x)$ replaced by $-u_g(y)$.

potential energy and kinetic energy in the final state and between the waves and the final steady state (Yacoby et al., 2021). As we shall shortly see, the effect of $\beta$ on the energy division is small for physically acceptable values of $b$.

Since Gill's seminal study (Gill, 1976) it became common practice to analyze the energetics of the adjustment problem by
calculating the energy conversion ratio [see, e.g., Grimshaw et al. (1998), Fang and Wu (2002), Yacoby et al. (2021)]:

$$\gamma = \frac{KE_g}{PE_0 - PE_g}$$

where $KE_g = \frac{1}{2} \int_{-L_y/2}^{L_y/2} u_g^2 \mathrm{d}y$ is the total kinetic energy of the geostrophic state, $PE_g = \frac{1}{2} \int_{-L_y/2}^{L_y/2} \eta_g^2 \mathrm{d}y$ is the total potential energy of the geostrophic state, and $PE_0 = \frac{1}{2} \int_{-L_y/2}^{L_y/2} \eta_0^2 \mathrm{d}y$ is the total potential energy of the initial state. Note that the total kinetic and potential energies are defined as the integral of the corresponding local values over the entire channel width $L_y$.
Figure 2 shows $\gamma$ as a function of $b$ for $L_y = 60$ (blue line and dots) and for $L_y = 4$ (red line and dots). The value of $\gamma$ is calculated in two ways (one shown by lines, the other by dots) that differ by the methods used for calculating the geostrophic state, $\eta_g$ and $u_g$. The first method is that employed in figure 1, i.e., it solves equations (30) and (32) using Matlab's bvp4c routine. Prior to the calculation of the integral of $\frac{1}{2}\eta_g^2$ and $\frac{1}{2}u_g^2$ over the entire channel width (to calculate $PE_g$ and $KE_g$) the values of $\eta_g$ and $u_g$ are spline interpolated. The values of $\gamma$ calculated in this way are shown by the blue ($L_y = 60$) and red





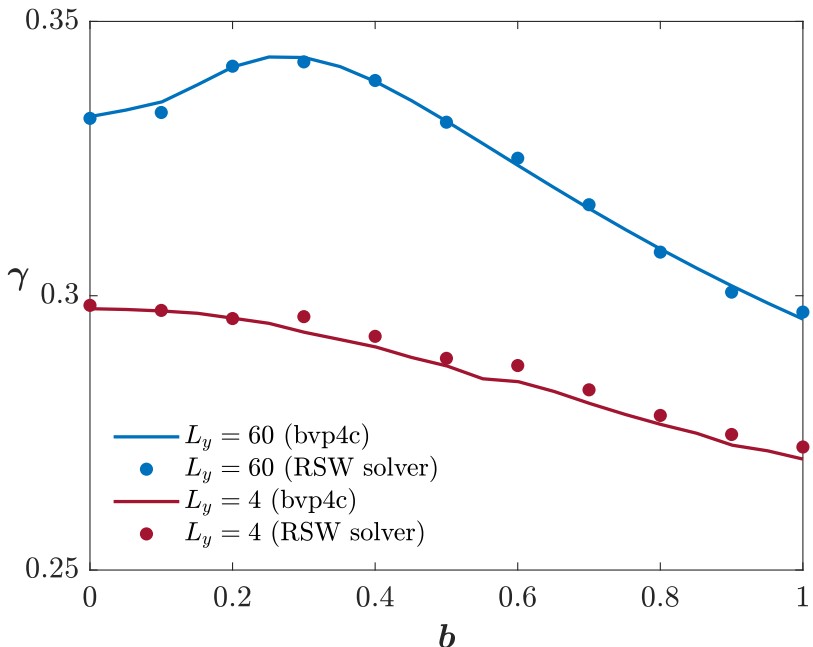

**Figure 2.** The energy conversion ratio, $\gamma$, as a function of $b$ for $L_y = 60$ (blue) and for $L_y = 4$ (red). Lines: the geostrophic state was calculated using bvp4c, as in figure 1; Dots: the geostrophic state was calculated using the RSW solver by taking the average of $\eta$ and $u$, as detailed in the text.

($L_y = 4$) lines. In the second method $\eta$, $u$ and $v$ are simulated using the RSW solver described in §2.2 (simulation **A**) and the simulated $\eta$ and $u$ values are averaged over many wave-periods. The averages were calculated between $t = 0$ and $t = 200$ with intervals of 0.1 time-units (i.e., intervals of $100\Delta t$, see table 1). Figure 2 shows that $\gamma$ varies with $b$ by no more than 10% compared to the $f$-plane values noted along the $b = 0$ ordinate; In a narrow channel ($L_y = 4$, red line and dots) the decrease in $\gamma(b)$ is monotonic, while in a wide channel ($L_y = 60$, blue line and dots) $\gamma$ has a local maximum of $\approx 0.343$ at $b \approx 0.3$.

**4.3   Poincaré waves**

The analysis of the $v(y,t)$ field employs the EOF method that examines the field's spatial and temporal patterns of variability [see, e.g., Björnsson and Venegas (1997) and Eshel (2011, chapter 11)]. We apply the EOF analyses to the $v(y,t)$ field simulated by the RSW solver (simulation **B**). Henceforth we choose $b = 0.005$, a typical value in the ocean where $R_d \sim 30$ km at $\phi_0 = \pi/4$.

The solid lines in figure 3 show the first four EOFs of $v(y,t)$ in a wide channel ($L_y = 60$) for $b = 0.005$. Each EOF mode can be associated with a different wave mode by examining its number of nodal points i.e. EOFs 1-4 are associated with wave modes $n = 3, 4, 6, 9$, respectively. The dashed lines show the Airy eigenfunctions given by equation (20) for $n = 3$ (blue), $n = 4$ (red), $n = 6$ (green) and $n = 9$ (purple) and the dotted lines - the corresponding harmonic modes given by equation (16) for





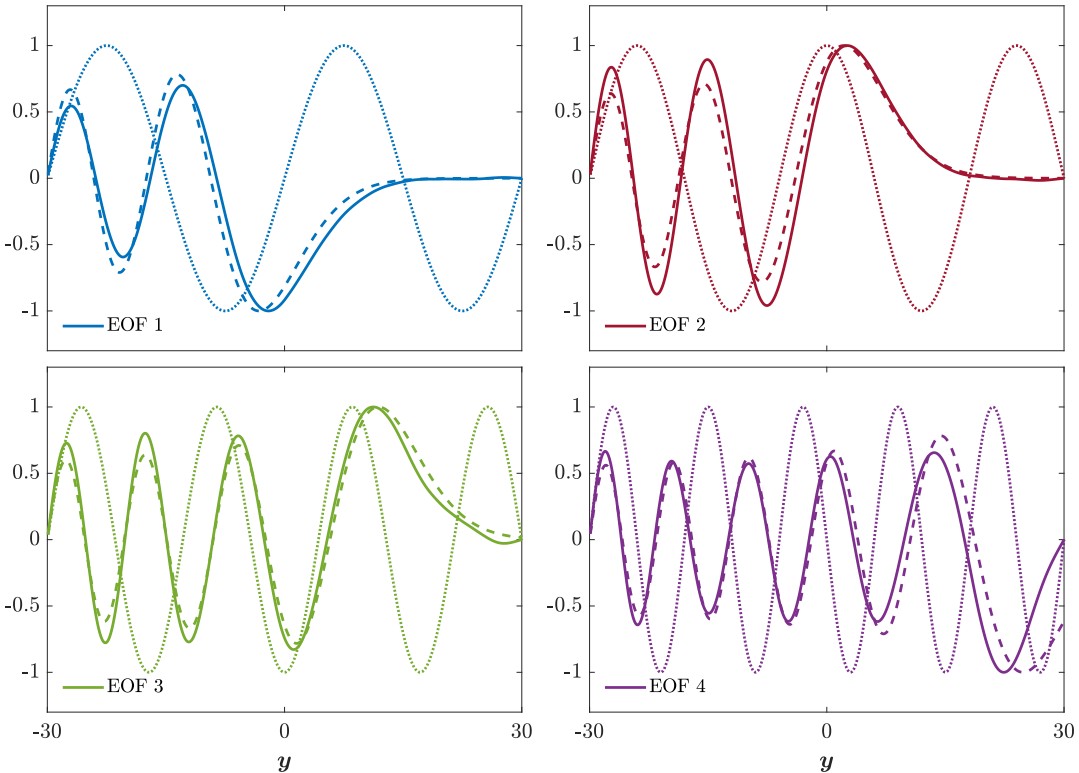

**Figure 3.** Solid lines: the first four EOFs of $v(y,t)$ in a wide channel. Dashed lines: the Airy eigenfunctions for $n = 3$ (blue), $n = 4$ (red), $n = 6$ (green) and $n = 9$ (purple). Dotted lines: the harmonic eigenfunctions for the same modes. In this, wide channel, case the EOFs resemble the Airy eigenfunctions.

the same mode numbers. All curves are normalized such that the maximal absolute value of their amplitude equals 1 and their derivative at $y = -L_y/2$ is positive.

Clearly, in a wide channel the EOFs are very similar to the Airy eigenfunctions. However, there is a significant deviation between EOF 4 (solid purple) and its associated Airy eigenfunction (dashed purple) near the northern wall, $y = L_y/2$. The reason for this deviation is the violation of condition (24) for large mode numbers i.e. when $\xi_n$ is too large. The fact that the EOFs match the Airy eigenfunctions accurately for $n = 3, 4, 6$ but not so for $n = 9$ results from the bound on the value of $Z^*$ (set to 2 as discussed at the end of §3.2). Specifically, condition (24) implies:

$$(2b)^{1/3} L_y - \xi_n > Z^*. \tag{36}$$

For the values of $L_y = 60$ and $b = 0.005$ used here, the LHS of equation (36) equals 2.886 for $n = 6$ ($\xi_6 = 10.040$), while for $n = 9$ ($\xi_9 = 12.829$) it equals 0.098. The results shown in figure 3 where EOF 3 matches nicely the $n = 6$ mode in the channel while EOF 4 matches the $n = 9$ mode only roughly, supports our choice of $Z^* = 2$ in (36). As could be expected, the mismatch between EOF 4 and and the $n = 9$ wave mode is maximal near the northern boundary while inward of it, the match is





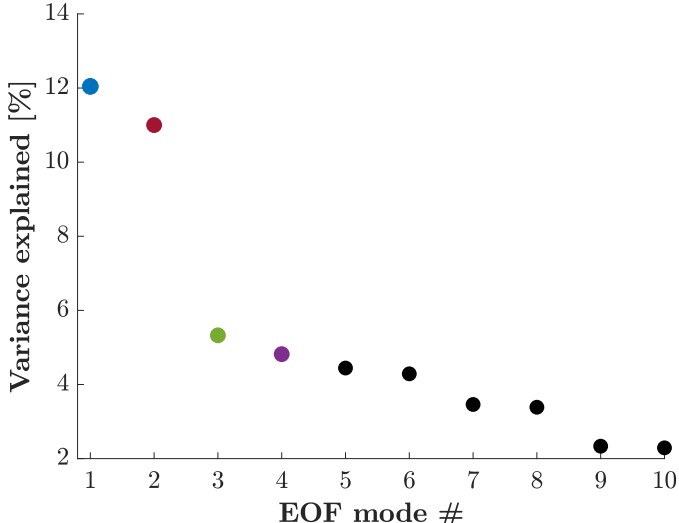

**Figure 4.** The variance explained by each of the first ten EOFs in a wide channel.

acceptable throughout most of the domain. The results also show that harmonic waves do not yield acceptable approximations to the calculated EOFs in the wide channel for any mode number.

In addition to the spatial structure of the EOF modes, the EOF analysis also yields the percentage of variance explained by each of the modes. The percentage of variance explained by each of the first ten first EOF modes in a wide channel is shown in figure 4. The first four EOFs explain $\sim 33\%$ of the variance while any mode higher than 10 accounts for less than $\sim 1\%$ of the variance.

The solid lines in figure 5 show the Fast Fourier Transform (FFT) of the first four Principal Components (PCs) obtained from the EOF analysis. The dashed vertical lines indicate the frequencies of the trapped-waves theory, equation (22) with $k = 0$, for $n = 3, 4, 6, 9$. The dotted vertical lines indicate the frequencies of the harmonic wave theory, equation (18a), for the same modes. As expected, the frequencies obtained from the FFT are much closer to the frequencies of trapped waves than to the frequencies of harmonic waves. Note that the dashed- and dotted-red lines overlap.

Figures 6-8 show the counterparts of figures 3-5 but for a narrow channel where $L_y = 4$ (and $b = 0.005$ as in a wide channel). Figure 6 shows that the first four EOFs of $v(y, t)$ (solid lines) are identical to the harmonic eigenfunctions, (16), with $n = 0, 2, 4, 6$ (dotted lines that overlap the solid lines). Clearly, the Airy eigenfunctions, shown by the dashed lines in figure 6, are irrelevant to the calculated eigenfunctions in a narrow channel. Figure 7 shows that over $96\%$ of the variability is explained by the first three EOFs which should be compared with a wide channel where the first three modes explain less than $30\%$ of the variability. We speculate that the EOF algorithm decomposes a signal more efficiently when the basis functions are harmonic. Figure 8 shows that the frequencies obtained from the FFT (solid lines) match those of harmonic modes (equation (18a) with $k = 0$) very well (dotted lines) while those of the trapped modes (equation (22) with $k = 0$) provide extremely poor estimates in a narrow channel (the 4 dashed vertical lines at $\omega$ slightly above 1).





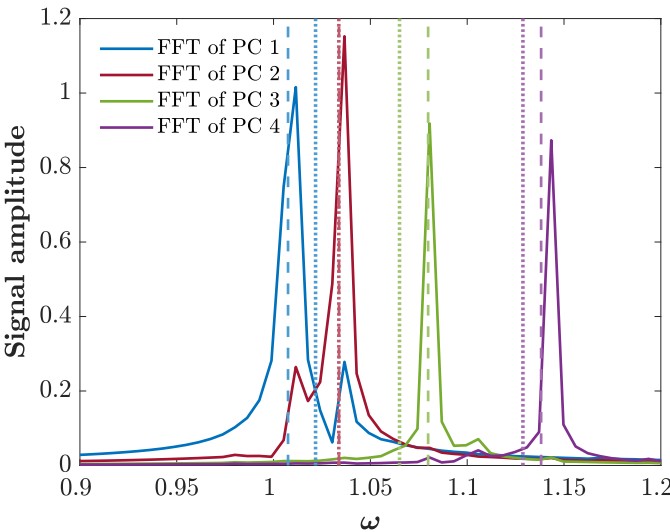

**Figure 5.** Solid lines: FFT of the first four PCs in a wide channel. Dashed lines: Trapped wave frequencies. Dotted lines: Harmonic wave frequencies.

## 5 Two-dimensional adjustment problem

Here we examine the adjustment problem in zonal channels when $\eta_0(x)$ is given by (11). Although $\eta_0$ depends only on x, the solution in this case is a function of both $x$ and $y$ since both the boundary conditions (8) and the Coriolis parameter [i.e., the term $(1 + by)$ in equations (4)-(5)] depend on $y$. Thus, the solutions in this case vary with $x$ as well as $y$.

### 285   5.1   The quasi-geostrophic state

When $\frac{\partial}{\partial x} \neq 0$ the dependence of the meridional variation of the Coriolis parameter generates Rossby waves. The frequency of Rossby waves is distinct from those of Poincaré and Kelvin waves, which causes the adjustment process to take place in two stages. In the first, relatively rapid stage $[t = \mathcal{O}(10)]$, Poincaré and Kelvin's waves propagate away from the initial disturbance, leaving behind them a quasi-geostrophic state. This stage resembles the geostrophic adjustment on the $f$-plane.
In the second stage, Rossby waves induce a slow westward motion of the initial disturbance and slowly deform the near geostrophic equilibrium established in the first stage (Blumen, 1972; Gill, 1982, section 11.11). Our simulations show that the second stage itself can be divided into two sub-stages where in the first sub-stage the near geostrophic state only drifts westward while its spatial structure is hardy changed while in the second sub-stage, wave dispersion becomes dominant, which alters the near geostrophic state.
Figure 9 shows the geostrophic steady state on the $f$-plane ($b = 0$) in wide ($L_y = 60$, left column) and narrow ($L_y = 4$, right column) channels. The steady solutions were calculated using the analytic expressions developed in subsection 5 of the Appendix in Yacoby et al. (2021) for long channels ($L_x \gg D$). In both channels the steady solutions vary with $y$ only near the boundaries so their variation with $y$ is less pronounced in wide channels as the solution there remains (nearly) independent of $y$





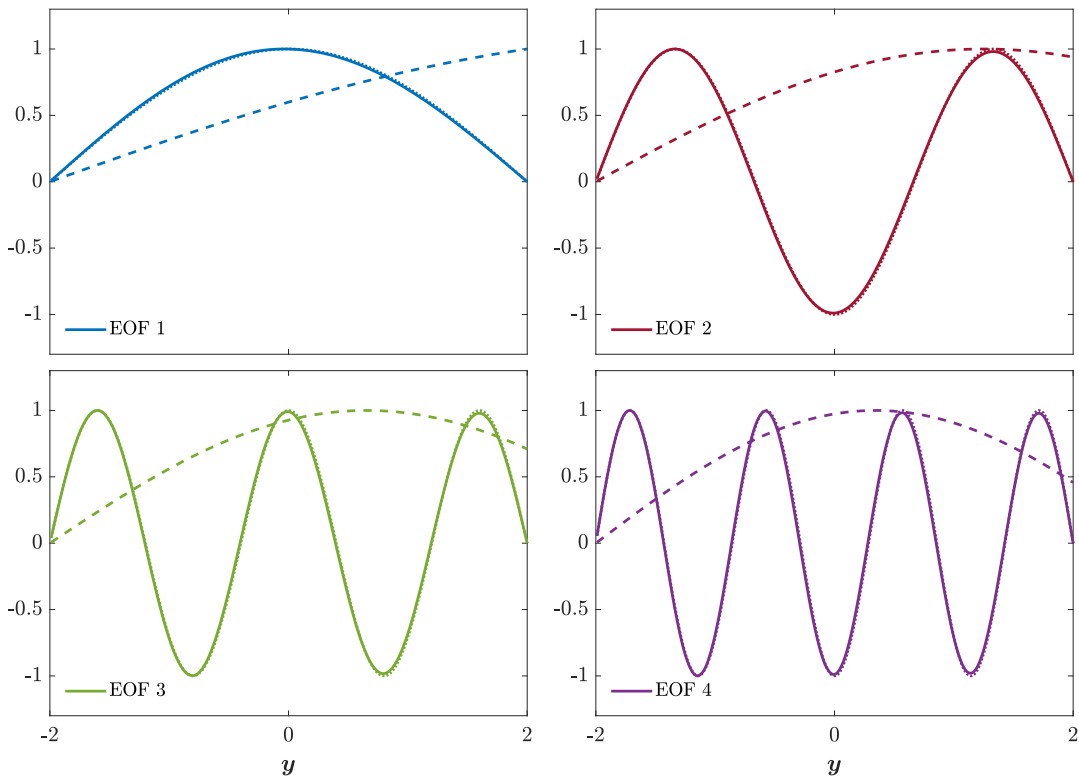

**Figure 6.** Solid lines: the first four EOFs of $v(y,t)$ in a narrow channel. Dashed lines: Airy eigenfunctions for $n=0$ (blue), $n=2$ (red), $n=4$ (green) and $n=6$ (purple). Dotted lines (indistinguishable from the solid lines): the harmonic eigenfunctions for the same modes. In contrast to a wide channel, in a narrow channel, the EOFs are identical to the harmonic eigenfunctions.

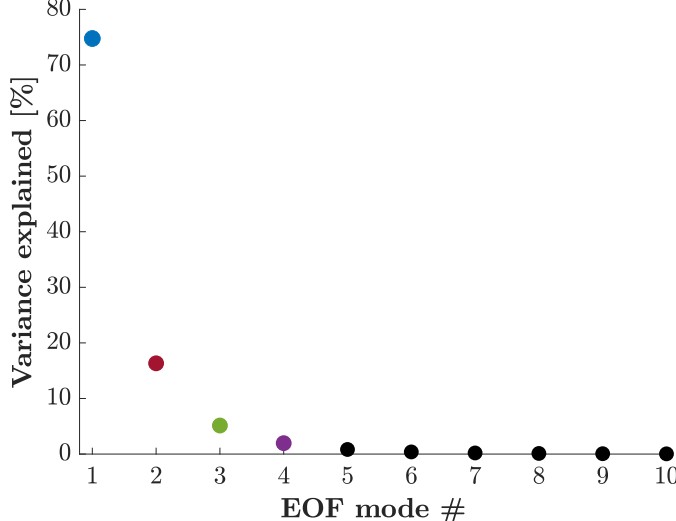

**Figure 7.** The variance explained by each of the first ten EOFs in a narrow channel.



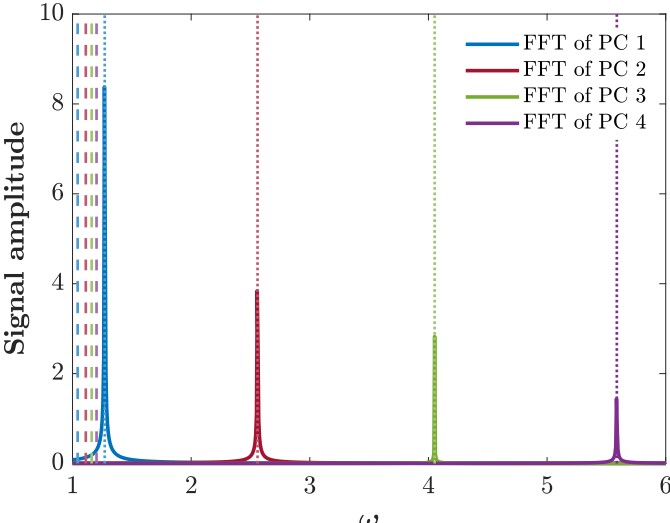

**Figure 8.** Solid lines: FFT of the first four PCs in a narrow channel. Dashed lines: Trapped wave frequencies. Dotted lines: Harmonic wave frequencies.

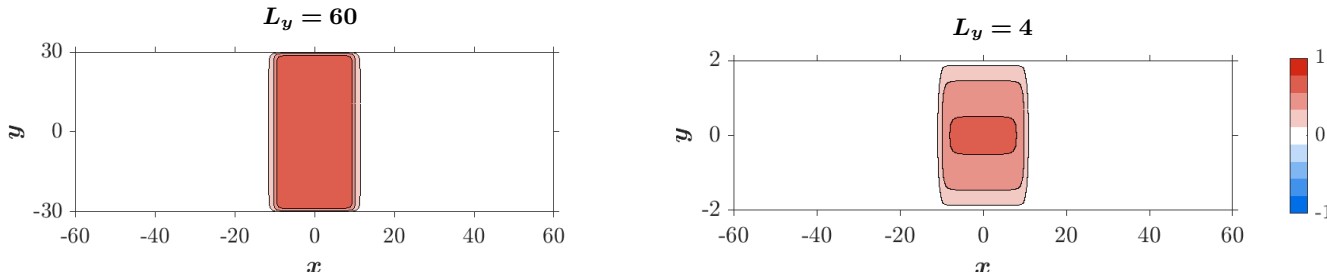

**Figure 9.** The geostrophic steady state on the $f$-plane in wide and narrow channels. Note the vastly different scales of the ordinates in the two panels

throughout most of the channel width (though the width of the boundary layer near the wall is not identical in the two channels, in a wide channel $\eta$ is nearly uniform over the $-25 < y < 25$ range). The steady state on the $f$-plane can be compared to the quasi-geostrophic state (i.e., the westward translating near steady state) on the $\beta$-plane illustrated in figure 10.

The temporal development of $\eta$ on the $\beta$-plane ($b = 0.005$) in wide and narrow channels was simulated with $L_x = 120$ so $D = L_x/12$ in equation (11) was set to 10 (simulation **C**). The second row in figure 10 shows the westward translation of the initial disturbance at $t = 1,000$ ($\sim 10^2$ days). During this (short) time the effect of wave dispersion is relatively small and the initial state only shifts westward by $\approx 15R_d$ while adjusting to a near geostrophic state (close to that shown in figure 9). Kelvin waves are also evident at this time near the meridional boundaries. In contrast, for $t \geq 7000$ (third to fifth rows in figure 10) the massive deformation due to wave dispersion is evident in the simulations. The simulations also highlight the fact that wave dispersion is more significant in a wide channel than in a narrow channel.







**Figure 10.** The temporal development of $\eta$ in wide and narrow channels. Note the emergence of Kelvin waves near the channel walls (most prominent in the narrow channel at $t = 1000$). The near geostrophic state is shifted westward at the speed of Rossby waves – harmonic in a narrow channel and trapped in a wide channel. The effect of dispersion increases with time. The vertical lines show the location of the leading edge based on harmonic wave theory [dotted lines, equation (37)] and based on trapped wave theory [dashed lines, equation (38)]. As expected, the harmonic wave theory yields more accurate estimates in a narrow channel while the trapped wave theory does so in a wide channel.



The speed of westward migration, i.e. the speed of the leading edge, $c_{\max}$, can be estimated from the phase speeds of long

harmonic and trapped Rossby waves i.e. by substituting $n = 0 = k$ in the dispersion relations. For harmonic waves (18b) implies:

$$c_{\max} = \frac{-b}{1 + \pi^2/L_y^2} \tag{37}$$

while for trapped Rossby waves (23) implies:

$$c_{\max} = \frac{-b}{1 + (2b)^{2/3}\xi_0 - bL_y}. \tag{38}$$

The vertical dotted and dashed vertical lines in figure 10 indicate the position of the leading edge using equations (37) and (38), respectively. Clearly, in a narrow channel the leading edge moves with the speed of harmonic Rossby wave while in a wide channel it moves with the speed of trapped Rossby wave. A comparison between the $\eta$ fields in the two channels at $t = 21,000$ (fifth row) shows that in a narrow channel the field resembles the $f$-plane steady state shown in figure 9 much more closely than in a wide channel i.e. the wave dispersion distorts the field more efficiently in a wide channel than in a narrow one.

From the left column of figure 10, it is evident that in a wide channel, where the trapped-wave theory applies, the southern part of the wavefront moves faster than the northern part, causing latitudinal tilting of the wavefront that increases with time. This phenomenon was established in prior numerical simulations of the $\beta$-plane, e.g., Sura et al. (2000) and Isachsen et al. (2007). The latitudinal tilting was heuristically attributed in these studies to the decrease with latitude of Rossby wave phase. Clearly, the harmonic wave theory cannot explain the latitudinal tilting since all long waves propagate at the same phase speed.

In the alternate, trapped-wave theory, the tilt of the wavefront is easily explained by the dispersion of the non-harmonic waves. In this theory the phase speed of trapped Rossby waves with $k = 0$ is:

$$c_{\text{ph}} = \frac{-b}{1 + (2b)^{2/3}\xi_n - bL_y}, \tag{39}$$

which decreases with $\xi_n$, i.e. the lower the wave mode, the faster the phase speed. Also, for higher wave modes the domain where the trapped wave oscillates and does not decay, extends farther away from the channel's southern boundary, causing

the wave peak to shift further northward. As a result, the southern part of the wavefront (where the low modes have $O(1)$ amplitude) moves faster than the northern part, which is dominated by high modes. This mechanism is illustrated in figure 11 where the sub-range in the channel where the amplitude of a particular mode is $O(1)$ is shown in panel (a) and the distance travelled by that mode in 7,000 time units is shown in panel (b) as a dot overlaid on the $\eta$ contours of the $t = 7,000$ panel on the left column in figure 10.

**5.2 Rossby waves**

As was done in the one-dimensional case studied in §4.3, we also use the EOF method to examine the spatial and temporal structures of the waves in the two-dimensional case. The analyses are performed on $v(x = -50, y, t)$ calculated by the MITgcm (simulation **D**).



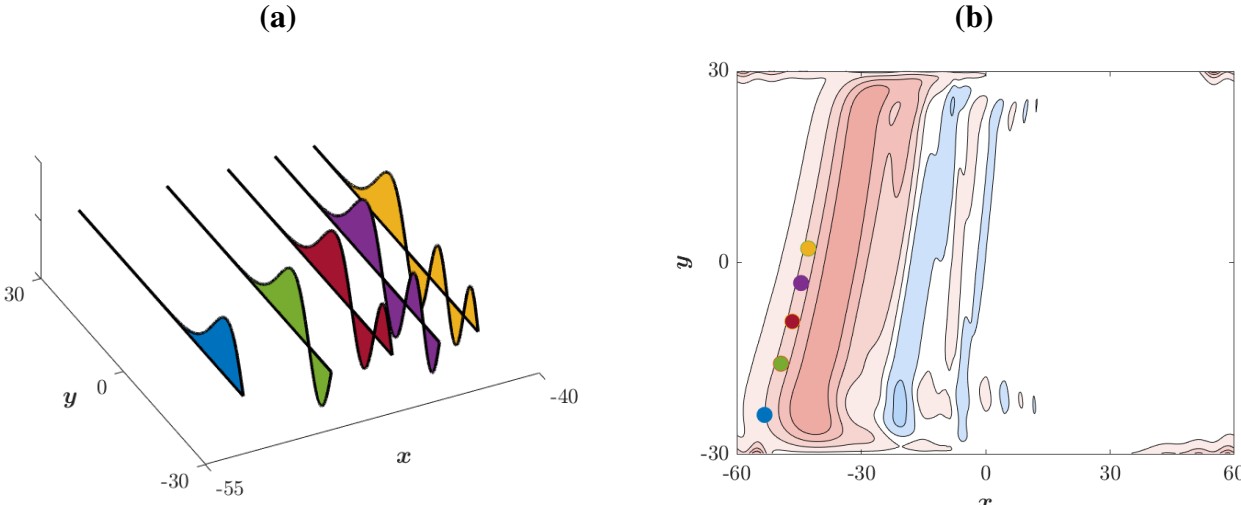

**Figure 11.** Dispersion of the Airy trapped waves in a wide channel. Panel (a): Structure of the first five trapped wave modes ($n = 0, 1, 2, 3, 4$) at $t = 7000$. The locations of the wave modes were calculated from the phase speed of trapped Rossby waves with $k = 0$, equation (39). The lower the wave mode, the faster the phase speed. As a result, the southern part of the wavefront moves faster than the northern part since low modes occupy only the former. Panel (b): Colored dots are the locations of the peaks of the Airy wave modes shown in panel (a) at $t = 7000$. The contours are $\eta$ at t=7000 (as in the third panel from the top in the left column of figure 10). The locations of the peaks are consistent with the numerical results (contours) and explain the slope in the wavefront.

Figure 12 shows the results of the analysis in a wide channel. The solid lines in panels (a)-(b) show the first two EOFs
of $v(x = -50, y, t)$. The dashed lines show the Airy eigenfunctions, (20), for $n = 0$ (blue) and $n = 2$ (red). The dotted lines show the harmonic eigenfunctions, (16), for the same wave modes. All curves are normalized as in figures 3 and 6. Similar to the results in the one-dimensional problem, in a wide channel the EOFs have the same structure as the Airy eigenfunctions. However, in the two-dimensional case, the differences between the EOF modes and the corresponding Airy eigenfunctions are more substantial than in the one-dimensional case (figure 3). We hypothesize that some of the differences between the EOFs

and the eigenfunctions result from a relatively low spatial and temporal resolution of simulation **D** (compared to simulation **B**). Clearly, the structures of harmonic modes approximate very poorly those of the EOF modes. Panel (c) shows the variance explained by each of the first ten EOF modes. The first two EOFs explain $47\%$ of the variance. The solid lines in panels (d)-(e) show the FFT of the first (blue) and second (red) PCs. The dashed-blue and dashed-red vertical lines show the frequencies of the $n = 0$ and the $n = 2$ trapped modes (respectively) given by equation (23), where $k = \pi(m + 1)/L_x$ for $m = 1, 3, 5, 7$. The

dotted-blue and dotted-red vertical lines show the harmonic frequencies given by equation (18b), for the same wave modes. Clearly, the frequencies obtained from the FFT are much closer to the trapped wave frequencies than to the harmonic wave frequencies.

Figure 13 shows the same information as figure 12 but for a narrow channel. Panels (a)-(b) show the first two EOFs that are identical to the harmonic eigenfunctions (16) with $n = 0$ and $n = 2$. As in the one-dimensional case, the Airy eigenfuntions





**Figure 12.** EOF analysis of Rossby waves in a wide channel. Panels (a)-(b), solid lines: the first two EOFs of $v(x = -50, y, t)$; dashed lines: Airy eigenfunctions for $n = 0$ (blue) and $n = 2$ (red); Dotted lines: the harmonic eigenfunctions for the same modes. Panel (c): The variance explained by each of the first ten EOFs. Panels (d)-(e), solid lines: FFT of the first two PCs. Dashed lines: the expected frequencies according to the trapped-waves theory [equation (23)]. Dotted lines: the expected frequencies according to the harmonic-waves theory [equation (18b)]

.





**Figure 13.** As in figure 12 but in a narrow channel.





provide a very poor approximation to the calculated structure of the EOF modes. Panel (c) shows that the first EOF explains nearly all the variance so each of the subsequent modes explains a minute fraction of the variance. Panels (d)-(e) show that the wave spectrum matches the spectrum of harmonic waves. Note that in panel (d) the trapped wave frequencies with $m \geq 3$ are outside the range of the abscissa.

### 5.3 Poincaré waves

Following the examination of Rossby waves we turn now to the examination of Poincaré waves. The calculations of the energy evolution, discussed in details in §5.4, reveal that the fraction of the initial (potential) energy converted to Poincaré wave energy during the adjustment process decreases with the length of the initial perturbation, $2D$. This is consistent with the results found on the $f$-plane in Yacoby et al. (2021). Hence, to increase the energy of Poincaré waves, we decrease the channel length, $L_x$, to 12 so $D = L_x/12 = 1$ instead of 10. The EOF analysis is performed on $v(x = 5, y, t)$ calculated by the RSW solver (simulation **E**).

The results of the EOF analysis in a wide channel presented in figure 14 clearly show that: (i) The numerically calculated EOFs of Poincaré waves are approximated by the Airy eigenfunctions more accurately than the EOFs of Rossby waves shown in figure 12 [panels (a)-(b)]; (ii) The first two EOFs explain $\sim 40\%$ of the variance, slightly less than the variance explained by the Rossby wave EOFs shown in figure 12 [Panel (c)]; (iii) The wave spectrum matches the spectrum corresponding to the Airy waves [panels (d)-(e)].

Figure 15 shows the same information as figure 14 but for a narrow channel. The differences between wide and narrow channels, evident from a comparison between between the two figures, are such that in a narrow channel: (i) The first two EOFs are identical to the harmonic eigenfunctions with $n = 0$ [Panel (a)] and $n = 2$ [Panel (b)]; (ii) The first EOF explains nearly all the variance [panel (c)]; (iii) The wave spectrum matches the spectrum of harmonic waves [panels (d)-(e)].

### 5.4 Energy

In the previous sections (§5.1-§5.3) we analyzed the results of simulations **C**, **D** ($L_x = 120$, $D = 10$) and **E** ($L_x = 12$, $D = 1$), each with both $L_y = 4$ and $L_y = 60$. In the present subsection, simulations **C**[1] and **E** are used to analyze the energy of Rossby waves and establish its relation to the energy of the steady state on the $f$-plane. Specifically, we compare the total (potential and kinetic) energy, $\frac{1}{2}\left(\eta^2 + u^2 + v^2\right)$, of Rossby waves on the $\beta$-plane to the total (potential and kinetic) energy of the steady state on the $f$-plane. To separate the low-frequency Rossby waves from the high-frequency Poincaré and Kelvin waves, we apply a 5[th]-order low pass Butterworth filter (in both forward and reverse directions). The cutoff frequency is set to $\omega = \pi/L_x$ (half the frequency of the Kelvin mode of the lowest frequency — $k = 2\pi/L_x$) which separates the frequencies of Rossby

---

[1] We prefer simulation **C** over simulation **D** because it has a better spatial and temporal resolution. The longer run-time of simulation **D** was irrelevant here since we analyzed the energy only for $t \leq 1000$ (see figure 16).







**Figure 14.** EOF analysis of Poincaré waves in a wide channel. Panels (a)-(b), solid lines: the first two EOFs of $v(x = 5, y, t)$; dashed lines: Airy eigenfunctions for $n = 0$ (blue) and $n = 2$ (red); Dotted lines: the harmonic eigenfunctions for the same modes. Panel (c): The variance explained by each of the first ten EOFs. Panels (d)-(e), solid lines: FFT of the first two PCs. Dashed lines: the expected frequencies according to the trapped-waves theory [equation (22) where $k = \pi(m + 1)/L_x$ with $m = 1, 3, 5, 7$]. Dotted lines: the expected frequencies according to the harmonic-waves theory [equation (18a) for the same wave modes].





**Figure 15.** As in figure 14 but in a narrow channel.





waves from the frequencies of Poincaré and Kelvin waves. However, since the integral of the surface initial height distribution,

$$\int\limits_{-L_y/2}^{L_y/2}\int\limits_{-L_x/2}^{L_x/2}\eta_0\mathrm{d}x\mathrm{d}y=2DL_y,$$

is not zero, Kelvin waves have a mean, time-independent, component $(\bar{\eta}_K,\bar{u}_K)$ in addition to the time-dependent high-frequency components. For small $b$ the mean component of Kelvin waves is approximated by:

$$\bar{\eta}_K=\frac{2D}{L_x}\Big(e^{-y-L_y/2}+e^{y-L_y/2}\Big)=\frac{4D}{L_x}e^{-L_y/2}\cosh\left(y\right),$$

$$\bar{u}_K=-\frac{\partial\bar{u}_K}{\partial y}=-\frac{4D}{L_x}e^{-L_y/2}\sinh\left(y\right).$$

To separate Rossby waves from the time-independent component of Kelvin waves, we subtract $\bar{\eta}_K$ and $\bar{u}_K$ from $\eta$ and $u$ after
applying the low pass filter.

The total energy of the steady state obtained on the $f$-plane was calculated using the analytic expressions developed in subsection 5 of the Appendix in Yacoby et al. (2021) for long channels (see figure 9 above). The solid lines in figure 16 show the total energy of Rossby waves (i.e. the integral over the entire channel) as a function of time. The dashed lines show the total energy of the steady state on the $f$-plane. All curves are normalized on the initial energy $-\frac{1}{2}L_y\int_{-L_x/2}^{L_x/2}\eta_0(x)^2\mathrm{d}x=L_yD$.
The figure shows that within a relatively brief time ($t\approx10$ when $D=1$ and $t\approx60$ when $D=10$), the energy of the Rossby waves reaches the final value that equals the energy of the steady state on the $f$-plane. We hypothesize that the difference in the time it takes the energy of Rossby waves to its final value for $D=1$ (simulation **E**) and $D=10$ (simulation **C**) is partially due to the difference in the temporal resolution between simulation **E** ($\Delta t=10^{-3}$) and simulation **C** ($\Delta t=10^{-1}$). The results shown in figure 16 affirm that Rossby waves on the $\beta$-plane are the counterpart of the steady state on the $f$-plane. For given
$L_y$, the initial energy that transforms to Rossby (Poincaré) wave energy increases (decreases) with $D$.

## 6    Discussion and summary

This paper examines the process of geostrophic adjustment in periodic zonal channels on the $\beta$-plane, focusing on the effect of $\beta$ on the: (i) Geostrophic steady state; (ii) Waves' structure and spectrum; (iii) Energetics of the adjustment process, i.e., the energy conversion ratio, $\gamma$, and the energy of Rossby waves. The three issues are addressed in the three following subsections,
§6.1-§6.3. Each subsection begins with a brief summary followed by a discussion.

### 6.1    The effect of $\beta$ on the geostrophic steady-state

In the one-dimensional case, the step in $\eta_0(y)$ given by (10) parallels the domain's zonal walls, i.e., parallels the $x$-axis. Thus, Poincaré waves propagate in the meridional direction and upon reaching the zonal walls they are reflected back into the interior of the channel. Since the reflection takes place in both walls Poincaré waves never leave the domain so the geostrophic state
shown in figure 1 only represents the time-averaged solution over many wave periods and not a steady state that is actually

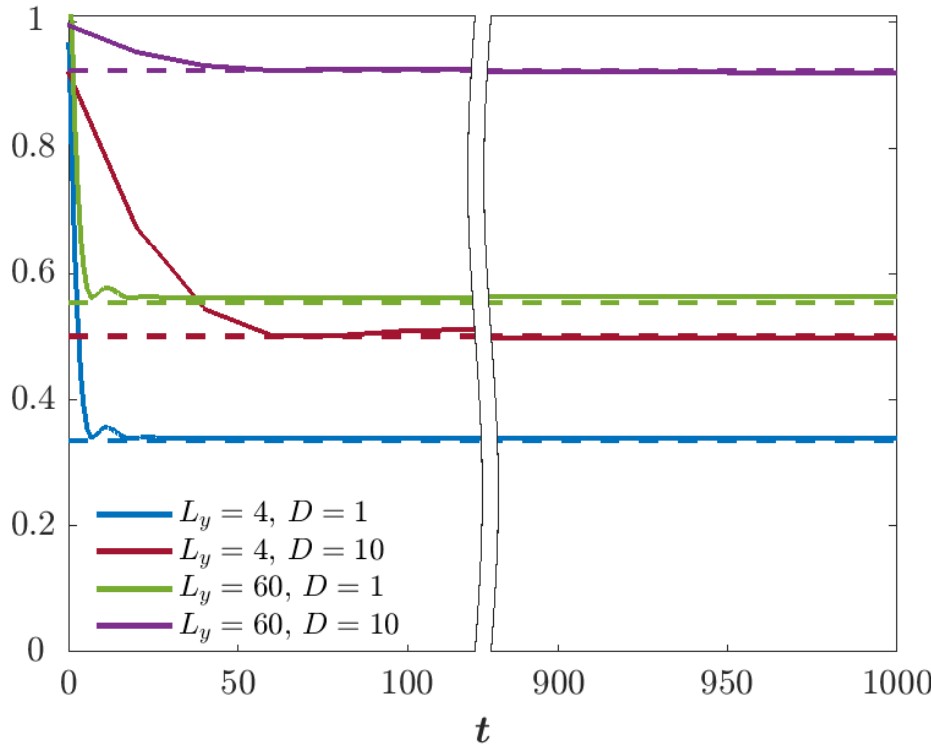

**Figure 16.** Solid lines: total (potential and kinetic) energy of Rossby waves as a function of time. Dashed lines: total energy of the steady state obtained on the $f$-plane. The curves were normalized by the initial energy, $L_y D$.

reached by the system at long times. As was shown in figure 1, the effect of $\beta$ on the geostrophic steady state is minor and its contribution is negligible for $b$-values that are typical for the ocean ($b = 0.005$, first row of figure 1) and atmosphere ($b = 0.05$, second row of figure 1). The independence of the steady state on $\beta$ even for $b = 0.05$ and $L_y = 60$ is surprising since for these values $bL_y/2 = 1.5 > 1$ (i.e. the $\beta y$ term exceeds $f_0$). Based on the calculations of the steady state in figure 1 we suggest that the $\beta$-effect should be quantified based on the (nearly exponential) distribution of $u_g(y)$ instead of the value of $bL_y$. According to this argument the $\beta$-effect is significant only when the Coriolis frequency, $1 + by$, vanishes not too far from the center of the channel i.e. at $y > -2$. Contrary to the naive approach based on a power series expansion of $f(y)$ near $y = 0$, the deviation of $1 + by$ from 1 does not affect the steady state. Our results suggest that the appropriate condition for the $\beta$-effect to be significant is not $bL_y/2 \sim 1$ but:

$$1 + by \leq 0 \quad \text{at} \quad y = -2$$

i.e., $b \geq 0.5$ irrespective of the value of $max(y) = L_y/2$. The significant effect of $\beta$ near the equator is demonstrated in Killworth (1991) on the equatorial $\beta$-plane and in Paldor and Dritschel (2021) on a sphere. In contrast, if the Coriolis parameter





vanishes far from the center of the channel, i.e., at $y \leq -2$, the effect of $\beta$ on the steady-state is insignificant. The reason for this is, of course, that at $y \leq -2$ the velocity, $u_g$, and with it the Coriolis force ($=fu_g$), are tiny according to figure 1.

In contrast to the one-dimensional case, in the two-dimensional case the effect of $\beta$ on the geostrophic state is significant also for $b = 0.005$ and also in narrow channels ($L_y = 4$). The effect originates from the formation of Rossby waves instead of the steady state and the slow westward translation of the quasi-geostrophic state by these waves. As time passes, the dispersion of Rossby waves increases, which alters the structure of the quasi-geostrophic state as shown in figure 10. As is evident in figures 10 and 11, in wide channels, the dispersion is more pronounced and causes latitudinal tilting of the front that increases

with time.

### 6.2    The effect of $\beta$ on the waves

For $b = 0.005$ we found that harmonic waves provide an accurate approximation for the waves in narrow channels while trapped waves provide an accurate approximation in wide channels, which agrees with the conclusion of Paldor and Sigalov (2008) and Gildor et al. (2016). Thus, the effect of $\beta$ on the spectrum and structure of the waves is significant in wide channels even

for small $b$. However, the definitions "wide" and "narrow" in this context have to be clarified. The transition from a narrow channel to a wide channel occurs when condition (24) is satisfied, which indicates that the definitions "narrow" and "wide" are not absolute but depend on: (i) the wave mode, $n$; the value of $b$; (iii) the value of $Z^*$ that should be "sufficiently large". The discussion of equation (36) is based on the choice of $Z^* = 2$. Thus, substituting $Z^* = 2$, $b = 0.005$ and $\xi_n = \xi_0 = 2.338$ implies that the condition for a channel to be "wide" is $L_y > 20.136$ (i.e. about 20 deformation radii or about $600km$ for

$R_d = 30km$). Furthermore, for higher wave modes the channel has to be even wider in order to satisfy this condition. These arguments justify our choice of $L_y = 4$ as an example of a narrow channel. On the other hand, $L_y = 60$ satisfies condition (24) even for $\xi_n = \xi_6 = 10.040$ which justifies our choice of $L_y = 60$ as an example of a wide channel.

It is worth noting that the satisfaction of condition (24) guarantees that the trapped wave theory will provide an accurate approximation but the violation of this condition does not imply that harmonic wave theory provides an accurate approximation.

An example where condition (24) is not satisfied and neither the harmonic wave theory nor the trapped wave theory provides an acceptable approximation is shown in the lower right panel of figure 3: EOF 4 (solid purple) does not perfectly match either the harmonic (dotted purple) or the Airy eigenfunction (dashed purple). Nevertheless, it resembles the Airy eigenfunction better than the harmonic eigenfunction.

To conclude the discussion in this subsection, we note that our examination of the waves focuses on the meridional velocity,

$v$, since this variable satisfies the boundary conditions (8) so the eigenvalue equation (12) is derived for this variable. Thus, the effect of $\beta$ on Kelvin waves is ignored here which is justified by the results reported in Paldor et al. (2007), who showed that in Kelvin waves the effect of $\beta$ is negligible.

### 6.3    The effect of $\beta$ on the energy

In the one-dimensional case figure 2 shows that the energy conversion ratio, $\gamma$, varies by no more than $10\%$ when $b$ increases

from 0 to 1. This near independence of $\gamma$ (that depends only on the initial state, $PE_0$, and the geostrophic state, $KE_g$ and



$PE_g$) on $\beta$ reflects the near independence of the steady state on $b$ (see figure 1). As discussed in §6.1, only near the equator (where $b \propto \cos(\phi_0)/\sin^2(\phi_0)$ is large) the effect of $\beta$ on the steady state is significant. As a result, near the equator $\gamma(b)$ is significantly different from $\gamma(b=0)$ – the value of $\gamma$ on the $f$-plane (results not shown).

In the two-dimensional case, we found that the zonal kinetic energy of Rossby waves (on the $\beta$-plane) is similar to the zonal kinetic energy of the steady state on the $f$-plane. This implies that the amount of initial energy that transforms into energy of the (high-frequency) Poincaré and Kelvin waves is nearly unaffected by $\beta$.

### 6.4 Additional initial and boundary conditions

The analysis presented above is limited to two initial height distributions, (10) and (11). While the transition from the trapped wave theory (in wide channels) to the harmonic wave theory (in narrow channels) is probably relevant to all initial conditions, some of the results presented above are expected to vary with the initial conditions, e.g., the distribution of energy between the different wave types (Rossby, Poincarè and Kelvin), the structure of the dominant EOFs and the variance explained by each of them. The present analysis is also limited to zonal periodic channels while other boundary conditions such as the introduction of meridional walls (i.e. a rectangular basin) require further analysis. A complete description of the geostrophic adjustment on the $\beta$-plane has to address the adjustment process under different initial and boundary conditions and this study is a first attempt in this direction.

### 6.5 Ocean response to a sudden wind stress

This study examines the adjustment process of an initially unbalanced ocean, where the waves are driven by an initial height disturbance. While progress on extending this particular problem to the mid-latitude $\beta$-plane has been limited, other ocean adjustment problems have been extensively studied on the $\beta$-plane in recent decades. An example of such an ocean adjustment problem is the ocean's response to a wind stress impulse that generates westward-moving Rossby waves that cause a thinning of the western boundary layer (see e.g., Anderson and Gill, 1975). The applicability of the trapped-waves theory to the wind-driven adjustment problem remains uncertain. Nonetheless, simulations of the forced LRSWE with no $x$-variations, i.e., equations (25)-(27) with wind forcing, suggest that trapped waves are indeed generated in this adjustment scenario (results not shown). Furthermore, the slope in the Rossby wavefront observed in the two-dimensional wind-driven adjustment problem (see e.g., Sura et al., 2000) agrees with the slope shown in figure 11 which was attributed to the dispersion of trapped waves but does not occur in harmonic waves.

### 6.6 Non-linear effects

This study focuses on linear theory on the $\beta$-plane. We note that the non-linear geostrophic adjustment on the $f$-plane is similar to the linear adjustment on the $\beta$-plane in that both occur in two stages. In the present study this is demonstrated in §5.1. In the non-linear adjustment on the $f$-plane, this was pointed out in Gill (1976, section 9), Hermann et al. (1989), Tomasson and Melville (1992), and Reznik et al. (2001). In the rapid first stage, the evolution is determined by the fast Poincaré waves



(that exist on both the $f$-plane and the $\beta$-plane). The second stage is much slower, and the mechanisms that drive it differ between the linear $\beta$-plane and the non-linear $f$-plane. While on the linear $\beta$-plane this stage is driven by Rossby waves, in the non-linear adjustment on the $f$-plane this stage is driven by advection of PV. We conclude by noting that Reznik et al.

490 (2001, section 5) remark (but do not develop a complete theory) on the non-linear geostrophic adjustment on the $\beta$-plane. This important issue is left for future work.

*Acknowledgements.* This research was supported by the ISF-NSFC Joint Research Program (Grant No. 2547/17 to H.G.).



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
