# Peer review of "Geostrophic adjustment on the mid-latitude $\beta$ -plane"

_EGUsphere, 2023_

## Author Comment (AC1)

**Response to Referee #1's comments on egusphere-2023-819**

by: I. Yacoby, H. Gildor, and N. Paldor

The Referee #1's comments are quoted below in blue and the authors' response are written in black

**Response to Referee 1**

My main point I would like to see addressed is the addition of a discussion early in the paper on the application of a channel for mid-latitude flows. There are no solid boundaries at mid-latitudes in the Atlantic or Pacific Oceans. For wide channels, the solutions trapped wave solutions depend on having a wall to support the waves. How would these waves be supported in the real ocean? Is choice of the zero crossing sufficient? Is there a zonal pressure gradient at y=-L/2?

In response to this comment we've added a new subsection ("6.7 applicability of the trapped wave theory to the ocean") to the discussion and this new subsection is referenced at the end of subsection 3.2.

Briefly, the answer to the first question is yes – the only requirement is that no water flows across the boundaries. As for the second question – Eq. (4) with $v = 0$ implies that, in the 2D case, at either of the 2 walls $\frac{\partial \eta}{\partial x} = -\frac{\partial u}{\partial t}$ at $y = \pm L_y/2$, so there is a (transient) zonal pressure gradient at the walls that obeys the same non-rotating dynamics (F=ma) in the zonal direction as Kelvin waves.

**Detailed comments:**

line 20: understanding
Done, thank you.

39: A brief statement of how the waves are trapped would be helpful.
As requested, a brief statement has been added to this paragraph.

Table 1: define what runs A-E are, either in the table or the caption.
A definition has been added to the caption.

132: What is lost when you neglect -2by and in what limit is this valid? You calculate solutions for by=O(1).
Neglecting the $-2by$ term yields harmonic solutions that are symmetric about $y = 0$ so the inherent asymmetry about $y = 0$ on the $\beta$-plane is lost.
The validity of the harmonic wave theory is a delicate and more general issue since condition (24) characterizes only the validity of the Airy wave theory but does not limit the validity of the harmonic wave theory. However, though analytic limitations of the validity of the harmonic wave theory do not exist, their validity can be examined numerically (see Paldor and Sigalov ,2008; and especially their Fig. 3). These numerical solutions verify that (at least for $R_d = 30km$) the harmonic theory yields good approximations of the first eigenvalue only when $L_y < 20$. This is a very subtle and important issue that deserves a more general analysis and cannot be fully addressed in passing while examining the geostrophic adjustment.

146: This is a confusing way to state that the Bi term must be zero in order for the solutions to be bounded.
The sentence has been rewritten.

Eqn (24) Can you provide some physical interpretation of this condition and the role of b and E?
Since Eq. (12) is a typical Schrödinger equation, $E$ can be interpreted as its energy level and $-2by$ as its potential (with $b$ – its only free parameter). A note has been added to the manuscript following Eq.

(12).
The condition on $L_y$ can be better understood by examining the dimensional form of Eq. (24), namely:

$$L_y > (2 + \xi_n)L^*$$

where:

$$L^* = \left(\frac{2\beta}{f_0 R_d^2}\right)^{-\frac{1}{3}}$$

is a new length scale that involves all parameters of the RSWE. The condition above implies that the ratio between the channel width $L_y$ and the new length scale, $L^*$, has to exceed the number $\xi_n + 2$ (which depends on the wave mode $n$). A more detailed examination of the upper and lower bounds of the two wave theories should be included in a dedicated study on the subject.

179: Mention the continuity equation (27) in going from (28) to (29).
Done.

183: Potential vorticity must also be conserved. What is the physical meaning of conservation of vorticity gradient?
Right. Thank you for pointing this out. In the revised version $q$ is defined as the perturbation potential vorticity (in Eq. (7)). The total potential vorticity, $Q$, as well as the relation between $q$ and $Q$ are now presented following Eq. (30). The new version also highlights the difference – $q$ is conserved locally only the $f$-plane while $Q$ is conserved Lagrangianly on both the $f$-plane and the $\beta$-plane.
No new physical meaning is proposed for this new conserved quantity. We'd appreciate any insight the readers of our paper can offer. It is worth mentioning that, in contrast to the conservation of $Q$, the conservation of the meridional gradient of $q$ is not general but holds only when $\partial/\partial x = 0$.

198: missing =0 on the du/dt term.
Done, thank you.

205: Please provide justification for the d(eta)/dy =0 boundary condition.
This boundary condition follows from the steady geostrophic relation $\frac{\partial \eta_g}{\partial y} = -(1+by)u_g$ since $u_g(\pm L_y/2 = 0) = 0$. The above justification has been added to the manuscript.

210: If the solution is symmetric about y=0, don't all solutions satisfy mass conservation?
(We assume the referee meant antisymmetric and not symmetric) In contrast to the $f$-plane, on the $\beta$-plane, the solutions are neither symmetric nor antisymmetric about $y = 0$. Yet, it is correct that since the **initial condition** $\eta_0(y) = \text{sgn}(y)$ is antisymmetric, mass conservation implies that the solutions must have vanishing integral of $\eta_g$. However, **numerically** the value of $a$ that zeros the integral can only be determined approximately following the calculation of the integral as follows. We first scan a finite range of $a$-values (based on the results presented in Fig. 1 the range was selected to be between $-1$ and $-0.3$ with intervals of 0.005) and choose the value of $a$ that yields the lowest value for the integral (which is practically zero). This clarification has been added to the manuscript.

Figure 4: So the variance is not well described by these modes since it takes so many.
The referee is right. For the particular discontinuous initial conditions used in this paper, the decomposition of the signal to the Airy functions in wide channels is less efficient than the decomposition of the signal to harmonic functions in narrow channels. However, the trapped waves theory explains the dominant modes (as well as additional modes that were not shown in the figure) in wide channels. In addition, the dominant EOFs and the variance explained by each of them may vary with the initial conditions, as mentioned in section 6.4. In particular, Gildor et al. (2016) show that when the RSW solver is initialized by a trapped wave with $n = 0$, this mode maintains its initial form (in wide channels), which implies that in this case nearly all the variance is well described by this particular mode. Moreover, a particular Airy mode that explains nearly all the variance can be obtained with arbitrary initial conditions as well by adding wind stress that changes periodically in time with a frequency that matches the frequency of this particular Airy mode (but the issue is not included in the paper).

323: The tilting of wave crests in many previous studies has been found in the absence of channel walls so it is most likely, at least in those cases, to be due to the variation in the long wave phase speed

with latitude. Linear Rossby wave theory has also been applied to forced variability in closed basins with close comparison to PE numerical solutions. The current discussion is too dismissive of the role of linear long Rossby waves at mid-latitudes.

Though, formally, the substitution of a variable $f(y) = f_0 + \beta y$ in the dispersion relation of Rossby waves yields the wave tilting as in the numerical simulations, the substitution is mathematically wrong. The reason is that the eigenvalue equation is derived by separating the variables of the LRSWE to $T(t)$ (usually assumed to be $e^{-ikCt}$) times $X(x)$ (usually assumed to be $e^{ikx}$) times $F(y)$ (that equals $e^{ily}$ for harmonic waves and $A_i(z(y))$ for trapped waves). Assuming that the phase speed $C$ is a function of $y$ violates the separation of variables and must be followed by the addition of a term proportional to $dC/dy$ to the differential equation that determines the eigenfunction (see an example in Sec. 3 of Garfinkel et al., 2017, QJ). Thus, the suggestion to add an explicit $y$-dependence to $C$ is inconsistent with the assumptions leading to the derivation of the explicit expression of $C$. This is a general perspective but insofar as Rossby waves on the $\beta$-plane are concerned, the basic issue is that all long harmonic waves (small $k$ and $l$) propagate at the same phase-speed while long trapped waves waves propagate at different speeds as is evident from the dependence on $n$ in equation (23) for $k \to 0$. Somehow, the addition of $-2by$ to the potential yields, in a mathematically consistent manner, wave properties (phase speed and amplitude structure) that agree with the numerical results. In the trapped wave theory the tilting results from the $y$-dependence of the amplitude while the phase speed (which is the eigenvalue of the equation) is constant.

345: Refer to Table 1 for model resolution.
Done.

Summary and Discussion: This section is very helpful and answered several questions that I had while reading the paper.
Great.

437: state again what Z* and zeta are.
Done.

Add a subsection 6.7 on the assumption of walls at +- Ly/2, which do not exist in the ocean or atmosphere.
The new subsection "6.7 The applicability of the trapped wave theory to the ocean" addresses this issue.

---

## Author Comment (AC2)

**Response to Referee #2's comments on egusphere-2023-819**

by: I. Yacoby, H. Gildor, and N. Paldor

The Referee #2's comments are quoted below in blue and the authors' response are written in black

**Response to Referee 2**

The specific domain used here is the zonal channel on the beta plane. This is arguably slightly restrictive but nevertheless useful. The presentation is generally clear. What I think would be useful would be a explicit demonstration of how the numerical results approach the standard and well-known f-plane results, as $\beta$ tends to zero. both for the time evolution and the final state. This would both provide confidence in the numerical scheme and provide insight into the effects of beta. Some f-plane results are provided, and but the physical patterns arenot shown except in figure 10 (which is showing something different).

Results on the $f$-plane ($\beta = 0$) are shown explicitly in figure 1 (dashed lines), figure 2 (the blue and red dots at $b = 0$), figure 9, and figure 16 (dashed lines) [but not in figure 10 as claimed]. Moreover, the harmonic wave theory (which neglects the term $-2by$) provides exact analytic solutions on the $f$-plane in the 1D case. Thus, the harmonic eigenfunctions (shown by the dotted lines in figures 3 and 6), as well as the harmonic spectrum (shown by the dotted lines in figures 5 and 8), are all $f$-plane results. Furthermore, the same authors have recently published a paper on the geostrophic adjustment on the $f$-plane (Yacoby et al, 2021; PoF), in which they employed the same initial conditions and similar channel configurations. Given this overlap, we believe that including additional results on the $f$-plane in the current manuscript would be superfluous. However, following this comment, an additional reference to the geostrophic adjustment on the $f$-plane has been added to the manuscript at the end of subsection 5.1.

Relatedly, potential vorticity is presumably conserved, and conserved pointwise in the linear problem. This is the basis for how the final state is calculated in the original Rossby problem. Some discussion of how PV enters into this problem and provides constraints on the evolution and final state would be welcome. Or if potential vorticity is not conserved or is somehow not relevant, some discussion of that would be useful.

This basic issue is now addressed in the paragraph following equation (30). Briefly, we derive the linearized local changed in $q$ (i.e. equation 7) directly from the governing equations and not from the nonlinear Lagrangian conservation of $Q$. Differentiating Equation (7) WRT $y$ yields the local conservation equation (28).